# A Survey of Helminths of Dogs in Rural and Urban Areas of Uzbekistan and the Zoonotic Risk to Human Population

**DOI:** 10.3390/pathogens11101085

**Published:** 2022-09-23

**Authors:** Alisher Safarov, Andrei D. Mihalca, Gab-Man Park, Firuza Akramova, Angela M. Ionică, Otayorjon Abdinabiev, Georgiana Deak, Djalaliddin Azimov

**Affiliations:** 1State Committee of Veterinary and Livestock Development of the Republic of Uzbekistan, Tashkent 100123, Uzbekistan; 2Department of Parasitology and Parasitic Diseases, University of Agricultural Sciences and Veterinary Medicine of Cluj-Napoca, 400372 Cluj-Napoca, Romania; 3Department of Environmental Medical Biology, Catholic Kwandong University College of Medicine, Gangneung 25601, Korea; 4Institute of Zoology of the Academy of Sciences of the Republic of Uzbekistan, Tashkent 100053, Uzbekistan; 5Clinical Hospital of Infectious Diseases of Cluj-Napoca, 400348 Cluj-Napoca, Romania; 6Samarkand Institute of Veterinary Medicine, Samarkand 140103, Uzbekistan

**Keywords:** dogs, helminths, Uzbekistan, zoonosis, necropsy

## Abstract

Dogs are very popular pets that can be infected with a wide diversity of endo- and ectoparasites, some of which have zoonotic potential. The aim of the present study was to determine the diversity and prevalence of helminths in rural and urban dogs in Tashkent, Samarkand and Karakalpakstan regions of Uzbekistan. A total of 399 dogs from rural and urban areas were examined by necropsy between November 2016 and March 2022. All helminth species were morphologically identified. A total of 31 species belonging to the classes Trematoda (3), Cestoda (9), Nematoda (18) and Acanthocephala (1) were identified in 378 dogs (94.7%). Twenty-one species are indicated for the helminth fauna of urban dogs and 31 species for rural dog populations. From the 31 species of helminths identified 18 species are zoonotic and four of them (*Echinicoccus granulosus, Dipylidium cani-num, Toxocara canis, Dirofilaria repens*) have a significant epidemiological importance. The study showed that the prevalence and diversity of helminths in dogs in rural areas of Uzbekistan is higher than in urban dogs.

## 1. Introduction

Knowledge of the parasitic helminth fauna of companion animals is not only of deep theoretical interest but is also of great practical importance as these can undermine human health. Among companion animals, dogs represent the most common pet worldwide having very close contact with humans. In the last decade, the interaction between dogs and humans has significantly increased, and nowadays pet dogs are considered members of the family [1]. At the same time, dogs can be infected with a wide diversity of endo- and ectoparasites, some of them with zoonotic potential [2]. Humans can get infected with different parasitic pathogens through direct contact with animal hair, food, or water, contaminated with canine excreta, or by consumption of canids meat, which is a culinary habit in some regions of the world [3,4]. Another global health problem is the environmental contamination of public areas by dog feces [5,6]. Moderate to severe or even lethal infections in humans can be caused by common endoparasites of dogs, such as *Toxocara canis* and *Ancylostoma caninum*, or by other helminths like *Echinococcus* spp., *Diphyllobothrium latum* and *Trichinella* spp. [7]. Accurate taxonomic identification of the parasites is the first prerequisite for the successful implementation of control measures as one of the most important risk factors for human infections with parasites transmitted by dogs could be represented by the lack of effective anthelmintic treatment associated with the absence of parasitological surveys [8].

Several studies on the helminth fauna of dogs were carried out in Uzbekistan between 1950 and 1975 [9,10,11,12,13,14,15,16,17,18]. According to these, the total infection rate of dogs ranged between 83.5–94.5%, with a total of 41 species of helminths being reported—14 of them having a zoonotic potential, of which three species are particularly dangerous to humans and have important epidemiological significance: *E. granulosus, D. caninum* and *T. canis*. However, throughout the 47 years gap between these reports and the present day, the size and the structure of the dog populations have changed significantly in Uzbekistan, increasing to 2,5 million (stray dogs are not included), due to several factors including international travel of animals and humans, globalization, land use, and climate change, as well as modifications of social and demographic features. Thus, the assessment of the current structure and burden of endoparasites in both the rural and the urban dog populations has become imperative. Therefore, in the present study, we aimed to investigate the current status of canine helminthic infection in rural and urban areas of Uzbekistan, to implement activities for minimizing the risks to the human population.

## 2. Materials and Methods

### 2.1. Sample Collection

The study was conducted in three regions of Uzbekistan, having different climatic conditions: Tashkent—humid continental climate, Samarkand-Mediterranean climate, and Karakalpakstan-extreme continental climate. Samarkand is about 267 km southwest from Tashkent and Karakalpakstan is 807 km west of Tashkent (Figure 1).

Samples were collected between November 2016 and March 2022 from a total of 399 dogs (160 in rural, and 239 in urban areas), collected as roadkills or euthanized in public dog shelters - in Tashkent (n = 200), Samarkand (n = 80) and Karakalpakistan (n = 119). Dogs were classified into three age groups as puppies (0–6 months), young dogs (>6 months to 12 months), and adults (>12 months) as described by Bone, 1988 [19].

All the animals were tested for rabies at the Virology Laboratory of the Republican State Center for Diagnosis of Animal Diseases and Food Safety before the helminthological examination. The carcasses were stored −20 °C prior to the necropsy. The study was reviewed and approved by the center′s ethics committee. 

### 2.2. Parasitological Examination

The carcasses were examined at the Republican State Center for Diagnosis of Animal Diseases and Food Safety. The subcutaneous tissues, eyes, body cavities, and surface of the internal organs were visually inspected for the presence of helminths. Following visual inspection, all the organs were dissected and examined separately. When present, helminths were collected into labelled vials. Cestodes were fixed in an alcohol-formaldehyde-acetic acid (AFA) solution or 5% formaldehyde, while nematodes and trematodes were placed in 70% alcohol at 60 °C and then fixed in 5% formaldehyde; larval nematodes and trematodes were removed from cysts with the aid of preparation needles and fixed in hot 4% formaldehyde; acanthocephalan larvae were placed in distilled water for 24 h at 4 °C for the proboscis to evert, and then fixed in hot 4% formaldehyde. Species that could not be identified immediately were processed using the technique described by Meyer and Olsen (1988) [20]. The collected helminths were sent for identification to the General Laboratory of Parasitology of the Institute of Zoology of the Academy of Sciences of the Republic of Uzbekistan. 

Species identification of parasites was carried out in accordance with the keys and descriptions reported previously [21,22,23].

The statistical analysis was performed using EpiInfo 7 software (CDC, USA). The prevalence and 95% confidence interval (CI) were calculated for each species of parasite, both globally, and according to environment and region. The differences among groups were assessed by chi-square test and were considered significant at a *p* value <0.05.

## 3. Results

Of the 399 examined dogs, parasitic helminths were found in 378 animals (94.7%; 95% CI 92.09-96.53%). The morphological identification revealed the occurrence of 31 species of helminths belonging to 4 classes: Nematoda *(Capillaria plica, Dioctophyme renale, Trichuris vulpis, Strongyloides stercoralis, Ancylostoma caninum, Uncinaria stenocephala, Toxascaris leonina, Toxocara canis, Physaloptera praeputialis, P. sibirica, Gongylonema pulchrum, Rictullaria affinus, R. cahirensis, Spirocerca lupi, S. arctica, Crenosoma vulpis, Dirofilaria immitis, D. repens)*, Cestoda *(Diphyllobothrium latum, Dipylidium caninum, Joyeuxiella rossicum, Mesocestoides lineatus, Taenia hydatigena, T. pisiformis, T. multiceps, T. taeniaeformis, Echinococcus granulosus),* Trematoda *(Alaria alata, Plagiorchis elegans, Dicrocoelium dendriticum)*, and Acanthocephala *(Macracanthorynchus catulinus)*. Out of these, 20 were found both in rural and urban environments, while 11 species were present exclusively in rural dogs (Table 1). Furthermore, for most species, the prevalence was higher in rural dogs as compared to urban ones, both globally (Table 1), and according to region (Table 2). The highest overall prevalence for 22 species was recorded in the Karakalpakistan region, while the differences between regions were significant in 25 instances (Table 3). 

Also, according to the obtained results, the nematode species *Physaloptera praeputialis* and *Physaloptera sibirica*, were identified on the rural area of Tashkent for the first time (Figure 2).

Among the identified helminth species, *Toxascaris leonina* and *Toxocara canis* were found to be dominant in all studied regions of Uzbekistan (Table 2, Figure 3).

Helminth species: *D. latum, T. pisiformis, A. alata, P. elegans, D. dendriticum, S. stercoralis, Ph. sibirica, G. pulchrum* and *R. cahirensis* were not detected in dogs in urban areas of Uzbekistan (Table 2).

Among the three age groups, the highest prevalence was recorded in puppies, followed by young dogs and adult ones, but with no statistically significant differences (Table 4). The prevalence of infection in female dogs was significantly higher as compared to males (Table 4).

Compared to the area with Mediterranean climate, it was observed that the level of infection of dogs with helminths was significantly higher in areas with humid continental and extreme continental climates (Table 4).

According to the results, of the 31 species of helminths identified 18 species are zoonotic (Table 2) and 4 of them *(Echinicoccus granulosus, Dipylidium caninum, Toxocara canis, Dirofilaria repens)* are being of significant epidemiological importance.

## 4. Discussion

Based on studies by Krabbe (1879) [24] and Linstov (1886) [25], five species of parasitic worms were registered in dogs of Turkestan: *Taenia hydatigena, Multiceps multiceps, Dipylidium caninum, Toxocara canis* and *Spirocerca lupi.* Later, several researchers [26,27] were engaged in the study of the helminth fauna of dogs on the territory of Uzbekistan, and registered 22 species of parasitic worms belonging to the classes Cestoda, Acanthocephala and Nematoda [28]. 

The present results on the fauna of helminths of dogs of rural populations partially confirm the data of earlier studies of 32 species of parasitic worms [1,3,10], which were summarized by Sultanov et al. (1975). *Toxocara canis, Toxascaris leonina, T. hydatigena* and *E. granulosus* were the most prevalent species, reaching up to 93.73%, 88.72%, 68.67%, and 63.91% prevalence respectively. Human disease by *Toxocara*, the most common parasite in dogs from Uzbekistan, occurs after accidental ingestion of the infective stages (eggs or larvae) and it is manifested in several syndromes like visceral and ocular larva migrans, covert toxocariasis and neurotoxocariasis [29]. In dogs from urban areas, the prevalence of this zoonotic nematode reached 98.74% and can represent a severe risk to humans. In Kazakhstan, another country in Central Asia, a very comprehensive study was done on the zoonotic risk of the population to several important helminth infections. The results showed a relatively high infection of the human population with *Echinococcus* spp. (more than 20 persons infected), *Toxocara* spp. (349 individuals, 11%) and *Toxoplasma gondii* (504 individuals, 16%) [30]. Interestingly, in the same country, the prevalence of these zoonotic helminths in dogs is lower than in dogs from Uzbekistan [30], highlighting the importance of large parasitological screening of dogs and humans. Human parasitological surveys are very limited in Uzbekistan, but infection with human cystic echinococcosis is considered to be higher than the official numbers [31]. Moreover, in recent years, cases of human infection with *Dirofilaria repens* species were recorded for the first time in Uzbekistan [32]. This situation indicates that the risk of zoonotic helminths in dogs has expanded.

Interestingly, several species of parasitic helminths: *Taenia ovis, Thominx aexophylus, Brachylaemus* sp., *Dracunculus medinensis, Filaroides osleri*; previously identified [9,11] were not detected in this study.

The frequency of helminth infections in dogs in rural areas of Uzbekistan is higher than in urban dogs. Similar results were obtained in parasitological surveys in the Czech Republic [33] and Hungary [34]. Dogs from rural areas could be more frequently infected due to the lack of prophylactic and metaphylactic treatments or it could be related to a higher density of free–ranging dogs in rural areas. In Uzbekistan, traditional husbandry is still practiced, and most dogs are used for security purposes. In contrast, in urban areas dogs are kept as pets and the majority have a good standard of health care [35]. 

While all noted species of helminths are common parasites of predatory mammals, including domestic dogs, both in rural and urban populations, there is a noticeable depletion of the helminth fauna diversity in urban dog populations. This may be related to the ecological characteristics of the current urban environment. Furthermore, the communities of helminths detected in rural dogs are uneven in different natural areas of Uzbekistan (Table 2). 

Climatic conditions in the humid zone are more favorable for the development and survival of the infective stages of some helminths in the environment, than in the arid region [36,37], which is in agreement with the present study.

The structure of the species that make up the canine parasitic worm fauna is heterogeneous in terms of their taxonomic structure and their relationship to other animal groups. Most species of dog helminths are ecologically related to vertebrates of all classes, which act as intermediate or reservoir hosts. The existing links between dog helminths and other groups of animals and humans from the point of view of veterinary and medical practice are of particular importance.

As integral components in the life of society, dog populations play a crucial role in the epidemiology of parasite species that affect the human population and domestic (productive) animals [32,33,34,35,36,37,38]. 

Given the high prevalence of zoonotic nematodes in dogs, it should be assumed that in modern conditions the problem of treating dog populations against parasitic diseases and protecting domestic animals and humans from them has not lost its relevance, on the contrary, its importance is growing.

The distribution of parasite communities depends on the ecological characteristics of the structure and functional features of rural and urban areas. In this context, the relationship between the parasitic fauna of the domestic dog and other vertebrates, including humans, deserves special attention. Constant surveys and control measures are mandatory for the prevention of infections in animals and humans.

## 5. Conclusions

More than 90% of tested dogs were infected with at least one parasite species based on morphological identification. No molecular work was done during the present study, which represents an important limitation. The high number of free–ranging dogs and stray dogs in urban areas can serve as a source of pathogens dangerous to humans. In this regard, dogs deserve special attention and epidemiological studies are important for determining the parasitological status of the population.

For this reason, we believe that it is necessary to educate the public about the rules of dog care at home and to further strengthen the control of stray dogs. Knowledge of the epidemiology of the main helminths in dogs is extremely important for conducting scientifically based implementation of anthelminthic measures. Prevention of parasite infections in dogs is an important element to secure human health.

## Figures and Tables

**Figure 1 pathogens-11-01085-f001:**
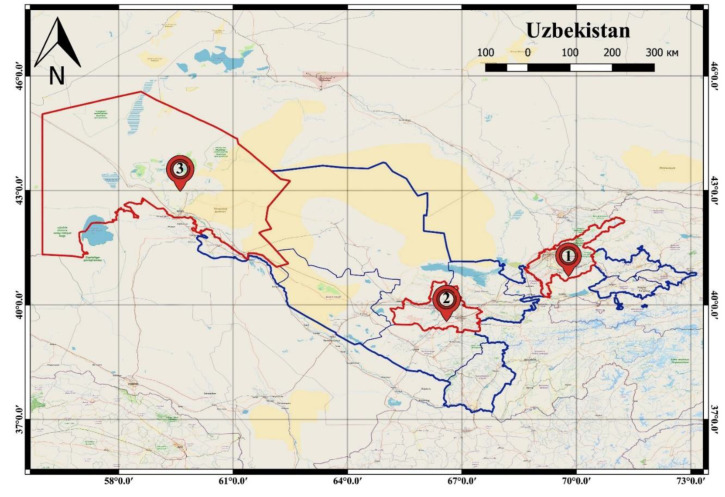
Survey regions in Uzbekistan (1—Tashkent; 2—Samarkand; 3—Karakalpakistan).

**Figure 2 pathogens-11-01085-f002:**
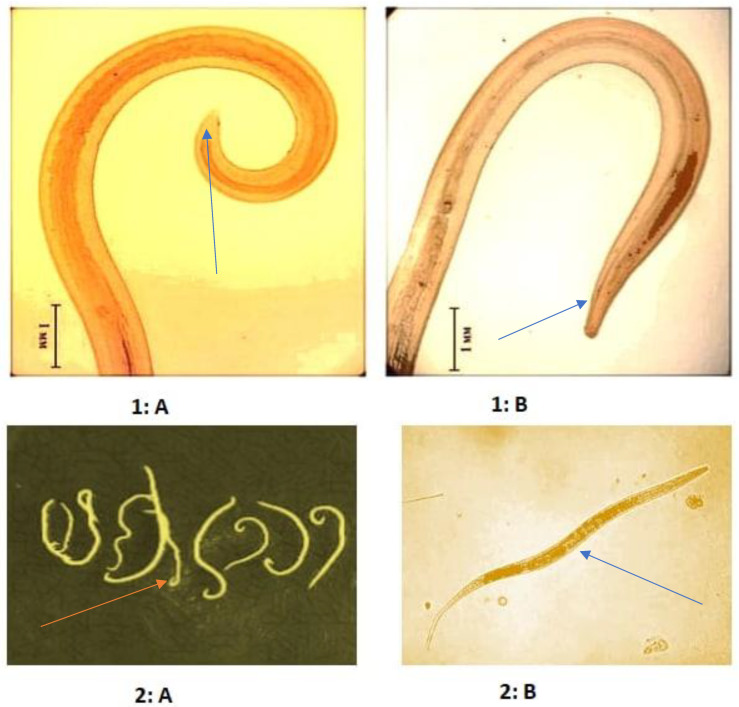
Morphological identification of the species *Physaloptera praeputialis* and *P. sibirica:*
**1: A** and **1: B**—*Physaloptera praeputialis* (Linstow, 1888), **1: A**—Posterior end; **1: B**—Anterior end; **2: A** and **2: B**—*P. sibirica* Petrow et Gorbunow, 1931, **2: A**—General appearance; **2: B**—Microscopic appearance.

**Figure 3 pathogens-11-01085-f003:**
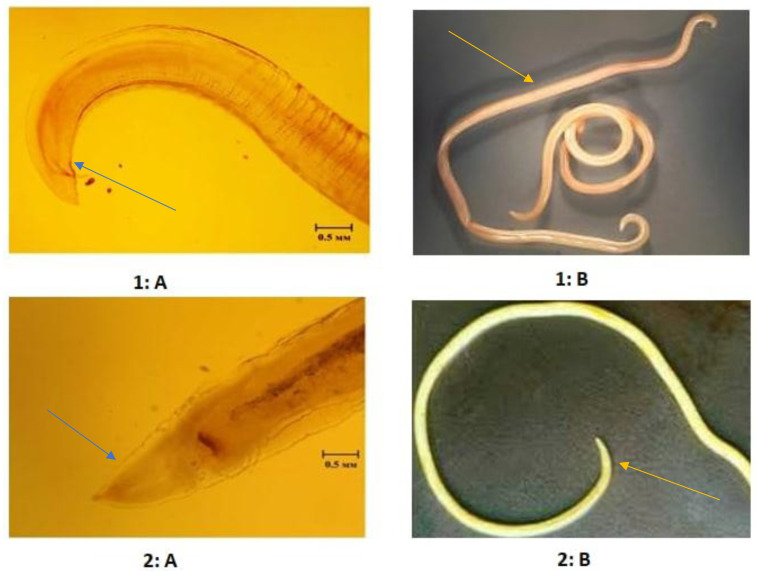
Morphological identification of *Toxocara canis* and *Toxascaris leonina.*
**1: A** and **1: B**—*Toxocara canis* (Werner, 1782), **1: A**—Posterior end; **1: B**—General appearance; **2: A** and **2: B**—*Toxascaris leonina* (Linstow, 1902), **2: A**—Anterior end; **2: B**—General appearance.

**Table 1 pathogens-11-01085-t001:** Prevalence of helminth infections of rural and urban dogs collected in the three-study area in Uzbekistan. Statistically significant values are highlighted in bold.

Class	Family	Species	TOTAL		Rural		Urban		χ^2^; d.f. = 1	*p*
			%	95% CI	%	95% CI	%	95% CI		
Cestoda	Diphyllobothridae	*D. latum*	31.58	27.21–36.3	78.75	71.59–84.81	0	0–1.53	**271.44**	**<0.0001**
	Dipylidiidae	*D. caninum*	51.38	46.48–56.25	58.13	50.08–65.87	46.86	40.4–53.4	**4.42**	**0.031**
		*J. rossicum*	38.85	34.19–43.71	44.38	36.53–52.43	35.15	29.1–41.56	3.05	0.08
	Mesocestoididae	*M. lineatus*	24.56	20.59–29.01	20.63	14.64–27.73	27.2	21.66–33.31	1.89	0.154
	Taeniidae	*T. hydatigena*	68.67	63.96–73.03	53.13	45.08–61.05	79.08	73.37–84.06	**28.81**	**<0.0001**
		*T. pisiformis*	32.58	28.17–37.33	81.25	74.33–86.98	0	0–1.53	**284.34**	**<0.0001**
		*T. multiceps*	46.37	41.53–51.27	63.13	55.15–70.61	35.15	29.1–41.56	**29.05**	**<0.0001**
		*T. taeniaeformis*	39.35	34.68–44.22	53.75	45.7–61.65	29.71	23.99–35.94	**22.21**	**<0.0001**
		*E. granulosus*	63.91	59.09–68.47	64.38	56.43–71.78	63.6	57.15–69.7	0.002	0.915
Trematoda	Diplostomidae	*A. alata*	16.29	12.99–20.23	40.63	32.94–48.66	0	0–1.53	**113**	**<0.0001**
	Plagiorchiidae	*P. elegans*	17.79	14.35–21.85	44.38	36.53–52.43	0	0–1.53	**125.99**	**<0.0001**
	Dicrocoeliidae	*D. dendriticum*	10.78	8.1–14.2	26.88	20.18–34.45	0	0–1.53	**69.22**	**<0.0001**
Acanthocephala	Oligacanthorinchidae	*M. catulinus*	12.03	9.19–15.59	17.5	11.95–24.29	8.37	5.19–12.63	**6.71**	**0.009**
Nematoda	Capillariidae	*C. plica*	31.58	27.21–36.3	46.88	38.95–54.92	21.34	16.32–27.08	**27.75**	**<0.0001**
	Trichocephalidae	*T. vulpis*	30.58	26.26–35.26	40	32.35–48.03	24.27	18.97–30.21	**10.44**	**0.001**
	Dioctophymidae	*D. renale*	39.85	35.16–44.73	69.38	61.61–76.41	20.08	15.2–25.73	**95.1**	**<0.0001**
	Strongyloididae	*S. stercoralis*	10.28	7.67–13.64	25.63	19.06–33.12	0	0–1.53	**65.5**	**<0.0001**
	Ancylostomidae	*A. caninum*	24.06	20.13–28.49	42.5	34.73–50.55	11.72	7.93–16.49	**48.03**	**<0.0001**
		*U. stenocephala*	36.84	32.26–41.68	45	37.14–53.05	31.38	25.55–37.68	**7.06**	**0.006**
	Crenosomatidae	*C. vulpis*	18.3	14.81–22.39	45.63	37.74–53.67	0	0–1.53	**130.42**	**<0.0001**
	Ascarididae	*T. leonina*	88.72	85.24–91.46	78.75	71.59–84.81	95.4	91.91–97.68	**24.9**	**<0.0001**
	Ascarididae	*T. canis*	93.73	90.91–95.72	86.25	79.93–91.18	98.74	96.38–99.74	**23.39**	**<0.0001**
	Spirocercidae	*S. lupi*	36.84	32.26–41.68	31.25	24.17–39.04	40.59	34.3–47.11	3.199	0.071
		*S. arctica*	17.04	13.67–21.04	42.5	34.73–50.55	0	0–1.53	**119.45**	**<0.0001**
	Physalopteridae	*Ph. praeputialis*	34.59	30.09–39.38	37.5	29.98–45.49	32.64	26.73–38.98	0.798	0.334
		*Ph. sibirica*	26.07	22–30.59	65	57.07–72.36	0	0–1.53	**206.75**	**<0.0001**
	Gongylonematidae	*G. pulchrum*	22.81	18.96–27.17	56.88	48.82–64.67	0	0–1.53	**172.87**	**<0.0001**
	Rictulariidae	*R. affinus*	33.33	28.89–38.1	39.38	31.75–47.4	29.29	23.6–35.5	**3.94**	**0.039**
		*R. cahirensis*	15.54	12.31–19.42	38.75	31.16–46.76	0	0–1.53	**106.71**	**<0.0001**
	Onchocercidae	*D. immitis*	11.53	8.76–15.04	12.5	7.81–18.64	10.88	7.23–15.53	0.11	0.634
		*D. repens*	8.02	5.74–11.1	9.38	5.34–14.99	7.11	4.2–11.14	0.39	0.53

**Table 2 pathogens-11-01085-t002:** Prevalence of helminth parasites isolated from the dogs in rural and urban areas in the Tashkent, Karakalakistan and Samarkand regions of Uzbekistan (*****—zoonotic species).

Species	Tashkent				Karakalpakistan				Samarkand			
	Rural		Urban		Rural		Urban		Rural		Urban	
	%	95% CI	%	95% CI	%	95% CI	%	95% CI	%	95% CI	%	95% CI
*D. latum **	57.5	40.89–72.96	0	0–2.28	80	67.67–89.22	0	0–6.06	91.67	81.61–97.24	0	0–16.84
*D. caninum **	52.5	36.13–68.49	41.88	34.13– 49.92	85	73.43–92.9	64.41	50.87–76.45	35	23.13– 48.4	35	15.39–59.22
*J. rossicum*	20	9.05– 35.65	33.13	25.9– 40.99	45	32.12–58.39	52.54	39.12–65.7	60	46.54–72.44	0	0–16.84
*M. lineatus*	7.5	1.57– 20.39	26.25	19.62– 33.78	35	23.13–48.4	18.64	9.69–30.91	15	7.1– 26.57	60	36.05–80.88
*T. hydatigena **	27.5	14.6– 43.89	80	72.96– 85.9	68.33	55.04–79.74	89.83	79.17–96.18	55	41.61–67.88	40	19.12–63.95
*T. pisiformis **	47.5	31.51–63.87	0	0–2.28	90	79.49–96.24	0	0–6.06	95	86.08–98.96	0	0–16.84
*T. multiceps **	52.5	36.13–68.49	27.5	20.75– 35.11	95	86.08–98.96	35.59	23.55–49.13	38.33	26.07–51.79	95	75.13–99.87
*T. taeniaeformis **	77.5	61.55–89.16	16.25	10.9– 22.9	63.33	49.9–75.41	49.15	35.89–62.5	28.33	17.45–41.44	80	56.34–94.27
*E. granulosus **	80	64.35–90.95	66.88	59.01– 74.1	81.67	69.56–90.48	45.76	32.72–59.25	36.67	24.59– 50.1	90	68.3– 98.77
*A. alata*	15	5.71– 29.84	0	0–2.28	63.33	49.9–75.41	0	0–6.06	35	23.13– 48.4	0	0–16.84
*P. elegans*	45	29.26–61.51	0	0–2.28	81.67	69.56–90.48	0	0–6.06	6.67	1.85– 16.2	0	0–16.84
*D. dendriticum*	15	5.71– 29.84	0	0–2.28	45	32.12–58.39	0	0–6.06	16.67	8.29– 28.52	0	0–16.84
*M. catulinus*	7.5	1.57– 20.39	10.63	6.31– 16.47	30	18.85–43.21	0	0–6.06	11.67	4.82– 22.57	15	3.21– 37.89
*C. plica*	22.5	10.84–38.45	3.13	1.02– 7.14	71.67	58.56–82.55	59.32	45.75–71.93	38.33	26.07–51.79	55	31.53–76.94
*T. vulpis **	27.5	14.6– 43.89	12.5	7.81– 18.64	55	41.61–67.88	47.46	34.3–60.88	33.33	21.69–46.69	50	27.2– 72.8
*D. renale **	72.5	56.11– 85.4	7.5	3.94– 12.73	75	62.14–85.28	32.2	20.62–45.64	61.67	48.21–73.93	85	62.11–96.79
*S. stercoralis*	65	48.32–79.37	0	0–2.28	18.33	9.52–30.44	0	0–6.06	6.67	1.85– 16.2	0	0–16.84
*A. caninum **	77.5	61.55–89.16	8.13	4.4– 13.49	35	23.13–48.4	22.03	12.29–34.73	26.67	16.07–39.66	10	1.23– 31.7
*U. stenocephala **	40	24.86–56.67	22.5	16.28– 29.76	65	51.6–76.87	45.76	32.72–59.25	28.33	17.45–41.44	60	36.05–80.88
*C. vulpis*	15	5.71– 29.84	0	0–2.28	66.67	53.31–78.31	0	0–6.06	45	32.12–58.39	0	0–16.84
*T. leonina **	47.5	31.51–63.87	98.75	95.66– 99.85	93.33	83.8–98.15	88.14	77.07–95.09	85	73.43– 92.9	90	68.3– 98.77
*T. canis **	52.5	36.13–68.49	99.38	96.57– 99.98	96.67	88.47–99.59	96.61	88.29–99.59	98.33	91.06–99.96	100	91.06– 100
*S. lupi **	40	24.86–56.67	48.75	40.78– 56.77	38.33	26.07–51.79	32.2	20.62–45.64	18.33	9.52– 30.44	0	0–16.84
*S. arctica **	52.5	36.13–68.49	0	0–2.28	55	41.61–67.88	0	0–6.06	23.33	13.38–36.04	0	0–16.84
*Ph. praeputialis*	75	58.8– 87.31	31.88	24.74– 39.7	50	36.81–63.19	38.98	26.55–52.56	0	0–5.96	20	5.73– 43.66
*Ph. sibirica*	67.5	50.87–81.43	0	0–2.28	80	67.67–89.22	0	0–6.06	48.33	35.23–61.61	0	0–16.84
*G. pulchrum **	47.5	31.51–63.87	0	0–2.28	88.33	77.43–95.18	0	0–6.06	31.67	20.26–44.96	0	0–16.84
*R. affinus*	82.5	67.22–92.66	43.75	35.93– 51.8	45	32.12–58.39	0	0–6.06	5	1.04– 13.92	0	0–16.84
*R. cahirensis*	45	29.26–61.51	0	0–2.28	51.67	38.39–64.77	0	0–6.06	21.67	12.07– 34.2	0	0–16.84
*D. immitis **	27.5	14.6– 43.89	10.63	6.31– 16.47	15	7.1– 26.57	3.39	0.41–11.71	0	0–5.96	35	15.39–59.22
*D. repens **	7.5	1.57– 20.39	3.13	1.02– 7.14	18.33	9.52–30.44	13.56	6.04–24.98	1.67	0.04– 8.94	20	5.73– 43.66

**Table 3 pathogens-11-01085-t003:** Prevalence of helminth parasites isolated from dogs in the Tashkent, Karakalakistan and Samarkand regions of Uzbekistan. Statistically significant values are highlighted in bold.

Species	Tashkent		Karakalpakistan		Samarkand		χ^2^; d.f. = 2	*p*
	%	95% CI	%	95% CI	%	95% CI		
*D. latum*	11.5	7.43–16.75	40.34	31.45–49.72	68.75	57.41–78.65	**92.7**	**<0.0001**
*D. caninum*	44	37.01–51.17	74.79	66.01–82.3	35	24.67–46.48	**39.05**	**<0.0001**
*J. rossicum*	30.5	24.2–37.39	48.74	39.47–58.07	45	33.85–56.83	**12.04**	**0.002**
*M.lineatus*	22.5	16.91–28.92	26.89	19.18–35.79	26.25	17.04–37.29	0.93	0.628
*T. hydatigena*	69.5	62.61–75.8	78.99	70.57–85.92	51.25	39.81–62.59	**17.24**	**0.0002**
*T. pisiformis*	9.5	5.82–14.44	45.38	36.23–54.76	71.25	60.05–80.82	**111.83**	**<0.0001**
*T. multiceps*	32.5	26.06–39.47	65.55	56.28–74.02	52.5	41.02–63.79	**34.27**	**<0.0001**
*T. taeniaeformis*	28.5	22.36–35.29	56.3	46.91–65.37	41.25	30.35–52.82	**24.31**	**<0.0001**
*E.granulosus*	69.5	62.61–75.8	63.87	54.55–72.47	50	38.6–61.4	**9.42**	**0.009**
*A. alata*	3	1.11–6.42	31.93	23.69–41.1	26.25	17.04–37.29	**53.07**	**<0.0001**
*P. elegans*	9	5.42–13.85	41.18	32.24–50.57	5	1.38–12.31	**64**	**<0.0001**
*D. dendriticum*	3	1.11–6.42	22.69	15.52–31.27	12.5	6.16–21.79	**30.38**	**<0.0001**
*M. catulinus*	10	6.22–15.02	15.13	9.22–22.85	12.5	6.16–21.79	1.87	0.391
*C. plica*	7	3.88–11.47	65.55	56.28–74.02	42.5	31.51–54.06	**123.88**	**<0.0001**
*T. vulpis*	15.5	10.78–21.27	51.26	41.93–60.53	37.5	26.92–49.04	**47.2**	**<0.0001**
*D. renale*	20.5	15.13–26.77	53.78	44.41–62.96	67.5	56.11–77.55	**69.39**	**<0.0001**
*S. stercoralis*	13	8.67–18.47	9.24	4.71–15.94	5	1.38–12.31	4.16	0.124
*A. caninum*	22	16.46–28.39	28.57	20.67–37.57	22.5	13.91–33.21	1.89	0.387
*U. stenocephala*	26	20.07–32.66	55.46	46.07–64.57	36.25	25.79–47.76	**27.84**	**<0.0001**
*C. vulpis*	3	1.11–6.42	33.61	25.22–42.85	33.75	23.55–45.19	**62.76**	**<0.0001**
*T. leonina*	88.5	83.25–92.57	90.76	84.06–95.29	86.25	76.73–92.93	0.99	0.609
*T. canis*	90	84.98–93.78	96.64	91.62–99.08	98.75	93.23–99.97	**9.88**	**0.007**
*S. lupi*	47	39.92–54.17	35.29	26.76–44.58	13.75	7.07–23.27	**27.32**	**<0.001**
*S. arctica*	10.5	6.62–15.6	27.73	19.92–36.68	17.5	9.91–27.62	**15.68**	**0.0004**
*Ph. praeputialis*	40.5	33.63–47.65	44.54	35.43–53.93	5	1.38–12.31	**39.25**	**<0.0001**
*Ph. sibirica*	13.5	9.09–19.03	40.34	31.45–49.72	36.25	25.79–47.76	**33.26**	**<0.0001**
*G. pulchrum*	9.5	5.82–14.44	44.54	35.43–53.93	23.75	14.95–34.58	**52.07**	**<0.0001**
*R. affinus*	51.5	44.35–58.61	22.69	15.52–31.27	3.75	0.78–10.57	**67.27**	**<0.0001**
*R. cahirensis*	9	5.42–13.85	26.05	18.44–34.89	16.25	8.95–26.18	**16.56**	**0.0003**
*D. immitis*	14	9.51–19.59	9.24	4.71–15.94	8.75	3.59–17.2	2.41	0.299
*D. repens*	4	1.74–7.73	19	9.9–23.81	6.25	2.06–13.99	**14.9**	**0.0006**

**Table 4 pathogens-11-01085-t004:** Prevalence of helminths in dogs in relation to age, sex, and climate (n = 399).

Variable	Examined	Infected			Χ^2^; d.f.; p
		n	%	95% CI	
**Age (months)**					
**0**–**6**	62	61	98.39	91.34–99.96	Χ^2^ = 2.7; d.f. = 2; *p* = 0.259
>6–12	207	193	93.24	88.91–96.25	
>12	130	124	95.38	90.22–98.29	
**Sex**					
Male	218	197	90.37	85.65–93.94	Χ^2^ = 16.52; d.f. = 1; *p* < 0.0001
Female	181	181	100	97.98–100	
**Climate**					
Humid continental	200	195	97.5	94.26–99.18	Χ^2^ = 24.22; d.f. = 2; *p* < 0.0001
Mediterranean	80	67	83.75	73.82–91.05	
Extreme continental	119	116	97.48	92.91–99.48	

## Data Availability

All relevant data is enclosed within the manuscript.

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
