# Peer review of "A Survey of Helminths of Dogs in Rural and Urban Areas of Uzbekistan and the Zoonotic Risk to Human Population"

_pathogens, 2022, doi:10.3390/pathogens11101085_

Round 1
Reviewer 1 Report
Please use italic in latin genus/species names throughout the whole text in lines: 24-31, 113-120, 133-134, 139, 144-146, 161-162, 168-169, 171, 178-179, 185, 188-189.
Repace historical name Trichocephalus vulpis for Trichuris vulpis in the text and tables.
Line 26: T. pisiformis
Line 44: ...different parasitic pathogens...
Line 45: ...with canine excreta,...
Line 46: ...canids meat, which is a culinary habit in some regions....
Line 54: Please change because: "the lack of effective anthelmintic treatment and the absence of parasitological surveys" are not "the main risk factors for human infections with parasites transmitted by dogs"
Line 70: Uzbekistan, to implement activities for minimizing the risks...
Line 96: 600C
Table 2 and 3: Toxoascaris leonina
Line187: ...indicates that the risk of zoonotic helminths in dogs has expanded.
Line 217: Given the high prevalence of zoonotic nematodes in dogs, it.
Line 232: ... to teach the public the rules
Lines 234-236: improve. The example below.
Knowledge of the epidemiology of the main helminths in dogs is extremely important for conducting scientifically based implementation of anthelminthic measures. Prevention of parasite infections in dogs is an important element to secure human health.
Author Response
Response to Reviewer 1 Comments
Point 1: Please use italic in latin genus/species names throughout the whole text in lines: 24-31, 113-120, 133-134, 139, 144-146, 161-162, 168-169, 171, 178-179, 185, 188-189.
Response 1: Corrected
Point 2: Repace historical name Trichocephalus vulpis for Trichuris vulpis in the text and tables..
Response 2: Corrected
Point 3: Line 26: T. pisiformis
Response 3: Corrected
Point 4: Line 44: ...different parasitic pathogens...
Response 4: Corrected
Point 5: Line 45: ...with canine excreta,..
Response 5: Corrected
Point 6: Line 46: ...canids meat, which is a culinary habit in some regions....
Response 6: Corrected
Point 7: Line 54: Please change because: "the lack of effective anthelmintic treatment and the absence of parasitological surveys" are not "the main risk factors for human infections with parasites transmitted by dogs"
Response 7: This was rephrased.
Point 8: Line 70: Uzbekistan, to implement activities for minimizing the risks...
Response8: Corrected
Point 9: Line 96: 600C
Response9: Corrected
Point 10: Table 2 and 3: Toxoascaris leonina
Response10: Corrected
Point 11: Line187: ...indicates that the risk of zoonotic helminths in dogs has expanded.
Response11: Corrected
Point 12: Line 217: Given the high prevalence of zoonotic nematodes in dogs, it.
Response12: Corrected
Point 13: Line 232: ... to teach the public the rules
Response 13: Corrected
Point 14: Lines 234-236: improve. The example below.
Response 14: This was rephrased. Thank you!
Reviewer 2 Report
See attached pdf file

Author Response
Response to Reviewer 2 Comments
Point 1: Line 3: change ''in'' to ''to''
Response 1: Corrected
Point 2: Line 18: delete with, and change to " some of which ''have'' zoonotic potential
Response 2: Corrected
Point 3: Line 18: add ''."
Response 3: Corrected
Point 4: Line 21: state specific period, eg March 2016 to June 2022.
Response 4: Corrected
Point 5: Line 21: mention how the various helminth species were identified. Morphological keys? etc
Response 5: Corrected
Point 6: Line 24-31: All these are not essential here. Instead mention the zoonotic species and risk factors.
Response 6: Corrected
Point 7: Line 34: add- necropsy
Response 7: Corrected
Point 8: Line 43: change ''which'' to ''them''
Response8: Corrected
Point 9: Line 51: ---- and Trichinella spp.
Response9: Corrected
Point 10: Line 56: Do you mean there have been no parasitological surveys of dog helminths in the country since 1975? If not, then modify this statement.
Response10: Yes, it is right. There are no complex studies of the helminth fauna of dogs since 1975
Point 11: Line 62: After writing the full names once, you abbreviate subsequently eg, E. granuosus, D. caninum etc.
Response11: Corrected
Point 12: Line 64: indicate the direction of the change in dog population; increase or decrease, and the level of veterinary/health care and dog management in the country
Response12: Corrected, it has been shown that the number of dogs has increased.
Point 13: Line 80: see earlier comments
Response 13: Corrected
Point 14: Lines 96:
Response 14: Corrected
Point 15: Line 97: not correct spelling
Response 13: Corrected
Point 15: Lines 109: test not testing
Response 14: Corrected
Point 17: Line 113: be consistent; you used class earlier.
Response 13: Corrected
Point 18: Lines 127: Tables 1-3 are too crowded and makes it a bit clumpsy. Authors need to reduce the font size, use abbreviation in names such that each raw stand out separately. also authors are encouraged to check and corrcer the spelling of the parasite names throughout the manuscript. wrong spellings are found in several places.
Response 14: Corrected. The tables have been corrected, the font size has been reduced, and the names of species have been shortened. Technical errors in the species names have been fixed.
Point 19: Lines 109: test not testing
Response 14: Corrected
Point 20: Line 135 and 142: Authors did well by providing pictographs. However, it will good for them to point out (using arrows) the key diagnostic features that enabled them to confirm identity of the helminths. Same for figure 1.
Response 20: Corrected
Point 21: Line 144-146: use abbreviations
Response 21: Corrected
Point 22: Line 155: move this to the left, to appear under variables.
Response 22: Corrected
Point 23: Line 156: rephrase this section please
Response 23: Corrected, newly edited
Point 24: Line 194: I am not conversant with this term. please can the authors explain?
Response 24: “metaphylactic” - Mass medication of a group of animals, in advance of an expected outbreak of disease.
Point 25: Line 221
Response 25: Corrected
Point 26: Line 232: change ''teach'' to ''educate''
Response 26: Corrected
Point 27: Line 236: Authors should mention the limitation of the study. The parasite identification was based on the use of morphological keys which is not very accurate especially for species idenification. This should be highlighted.
Response 27: This was mentioned in the conclusion section.
Point 28: Line 263: not cited in the text
Response 28: Corrected. Excluded from the list of references
Point 29: Line 303: not in the text
Response 29: Corrected. Excluded from the list of references
Point 30: Line 305: not in the text
Response 30: Corrected. Excluded from the list of references
Point 31: Line 318: not cited in the MS
Response 31: Corrected. Excluded from the list of references
Point 32: Line 325: not cited in text
Response 32: Corrected. Excluded from the list of references
Round 2
Reviewer 2 Report
No comments to the authors